# Patient journey following lumbar spinal fusion surgery (FuJourn): A multicentre exploration of the immediate post-operative period using qualitative patient diaries

Alison Rushton[1,2]*, Feroz Jadhakhan[2], Annabel Masson[2], Victoria Athey[2], J. Bart Staal[3], Martin L. Verra[4], Andrew Emms[5], Michael Reddington[6], Ashley Cole[7], Paul C. Willems[8], Lorin Benneker[9], Nicola R. Heneghan[2], Andrew Soundy[10]

1 School of Physical Therapy, Faculty of Health Sciences, Western University, London, Ontario, Canada, 2 Centre of Precision Rehabilitation for Spinal Pain, School of Sport, Exercise and Rehabilitation Sciences, University of Birmingham, Birmingham, United Kingdom, 3 Radboud Institute for Health Sciences, IQ Healthcare, Radboud University Medical Centre, Nijmegen, The Netherlands, 4 Department of Physiotherapy, Bern University Hospital, Insel Group, Bern, Switzerland, 5 Department of Physiotherapy, The Royal Orthopaedic Hospital NHS Foundation Trust, Birmingham, United Kingdom, 6 Department of Physiotherapy, Sheffield Teaching Hospitals NHS Foundation Trust, Northern General Hospital, Sheffield, United Kingdom, 7 Department of Orthopaedics & Trauma, Sheffield Children's Hospital NHS Foundation Trust, Sheffield, United Kingdom, 8 Maastricht University Medical Centre, Maastricht, The Netherlands, 9 Department of Orthopaedic Surgery Inselspital, University of Bern, Bern, Switzerland, 10 School of Sport, Exercise and Rehabilitation Sciences, University of Birmingham, Edgbaston, Birmingham, United Kingdom

* arushto3@uwo.ca

**Data Availability Statement:** All data are included in the paper and supporting files.

## Abstract

The aim of this study was to capture and understand the immediate recovery journey of patients following lumbar spinal fusion surgery and explore the interacting constructs that shape their journey. A qualitative study using Interpretive Phenomenological Analysis (IPA) approach. A purposive sample of 43 adult patients (≥16 years) undergoing ≤4 level instrumented fusion for back and/or leg pain of degenerative cause, were recruited pre-surgery from 4 UK spinal surgery centres. Patients completed a weekly diary expressed in their own words for the first 4 weeks following surgery to capture their life as lived. Diary content was based on previous research findings and recorded progress, recovery, motivation, symptoms, medications, healthcare appointments, rehabilitation, positive/negative thoughts, and significant moments; comparing to the previous week. To maximise completion and data quality, diaries could be completed in paper form, word document, as online survey or as audio recording. Strategies to enhance diary adherence included a weekly prompt. A framework analysis for individual diaries and then across participants (deductive and inductive components) captured emergent themes. Trustworthiness was enhanced by strategies including reflexivity, attention to negative cases and use of critical co-investigators. Twenty-eight participants (15 female; n = 18 (64.3%) aged 45–64) contributed weekly diaries (12 withdrew post-surgery, 3 did not follow through with surgery). Adherence with diaries was 89.8%. Participants provided diverse and vivid descriptions of recovery experiences. Three distinct recovery trajectories were identified: meaningful recovery (engagement in physical and functional activities to return to functionality/mobility); progressive recovery (small but

**Funding:** This work was supported by the Chartered Society of Physiotherapy Charitable Trust Physiotherapy Research Foundation, grant number PRF-16-A21. https://www.csp.org.uk/about-csp/how-we-work/charitable-trust Authors receiving award: AR, JBS, MV, AE,MR, AS, AC, PW, LB, NR. The funder played no role in study design, data collection and analysis, decision to publish, or preparation of the manuscript.

**Competing interests:** The authors have declared that no competing interests exist.

meaningful improvement in physical ability with increasing confidence); and disruptive recovery (limited purpose for meaningful recovery). Important interacting constructs shaped participants' recovery including their pain experience and self-efficacy. This is the first account of immediate recovery trajectories from patients' perspectives. Recognition of a patient's trajectory may inform patient-centred recovery, follow-up and rehabilitation to improve patient outcomes.

## Introduction

Worldwide, low back pain (LBP) is the single most common cause of physical impairment [1]. It is generally acknowledged that LBP is associated with functional impairment, work related absenteeism, long-term disability and high health care utilisation [2]. In the UK LBP accounts for approximately 40% [3] of all sickness absences in the National Health Service (NHS) and costs the UK economy a staggering £12.4 billion annually [4]. A wide range of treatment options are available to manage LBP. Surgical intervention such as lumbar spinal fusion surgery (LSFS) is widely available for people with severe, disabling, chronic low back pain that does not respond to structured conservative treatment [5]. The cost of LSFS to the NHS is reported to be in excess of £26 million annually [6], and in the USA LSFS is responsible for the greatest cost of any hospital surgery with estimates >$40 billion annually [7].

Studies question the increasing use of LSFS for the treatment of lumbar degenerative disorders owing to inconclusive trial outcomes and questionable cost effectiveness [8–11]. NICE, following the analysis of 9 studies (both randomised and cohort) investigating the clinical effectiveness of spinal fusion versus a range of comparators (usual care, conservative intervention, different surgery and treatments) found low level evidence for LSFS overall, but modest benefit for pain, function and quality of life; again with low level evidence [5]. Foster et al [12] as part of the recent Lancet low back pain series concluded that there was insufficient evidence regarding the role of LSFS in acute low back pain (non-radicular with degenerative disc findings), and that its role was uncertain for the persistent low back pain population.

Evidence of long-term outcome following LSFS has also been recognised as limited [13], particularly as trial outcomes are characteristically not assessed long-term (beyond 2 years). Previous data from the Swedish National Spine Register identified that 25% of patients reported no change or worsened pain following LSFS, and at 1 year 40% of patients reported dissatisfaction regarding their outcomes [14]. Conversely, in a recent systematic review and meta-analysis of 25 prospective cohort studies (n = 1777), the clinical course of pain (both leg and back pain) and disability in patients following LSFS for degenerative disorders of the lumbar spine was analysed [15]. The results demonstrated that leg pain, back pain and disability outcomes improved significantly, immediately following LSFS (within first 4 weeks). Results also indicated that in the long-term (evaluation was only possible to 24 months as no data were available >24 months) leg pain outcomes might remain more reduced and for longer than back pain and disability [15].

A number of surgical options related to LSFS are available, with substantial regional variation in treatment strategies and surgical rates. This illustrates a poor level of consensus regarding prognostic factors and indications for surgery, as well as heterogeneity of the patient group, and poor understanding and use of available predictive tests [16–18]. Lack of high quality evidence on the benefit and harm of LSFS [5] and the abovementioned lack of consensus results in uncertainty about the use and appropriateness of surgical intervention. A stratified model informing on the anticipated benefit of LSFS therefore appears important for surgical decision making and for rehabilitation following surgery in order to optimise outcomes [19].

Noticeably, the patients' voice through qualitative research is virtually missing from the current debate, with minimal research investigating patients' experiences of LSFS. Abbott et al [20] have conducted the only qualitative research to date through embedding qualitative interviews following LSFS into a clinical trial. Their focus was to relate the patients' experiences of their back problem, recovery from LSFS and rehabilitation expectations to components of the International Classification of Functioning, Disability and Health (ICF) [21]. Through content analysis, the ICF was used to code the qualitative data, identifying that the issues relating to the ICF components of body functions and structures (sensory, psychological, neuro-musculo-skeletal and movement), activities and participation (family relationships, domestic activities, mobility, recreation/leisure, work) and environmental factors (analgesia, walking aids, family support, healthcare and social systems, employment services) were important to recovery. Abbott et al [20] concluded that to determine outcomes important to patients following LSFS, use of the ICF core set [22] was recommended, as the current scope of outcomes used is limited and does not capture patients' experiences. This may partly explain the current uncertainty around the role of LSFS. This study reinforces that the patient's experiences and perceptions are central to understanding the role of LSFS, and why understanding the patient journey is important to determine best practice. The current study focuses on the immediate 4 week period following surgery; where limited data suggest improvements in key outcomes of pain and disability but where further detail is limited.

## Aim

To capture and better understand patients' lived experiences in the immediate period following LSFS.

## Objectives

1. To capture a weekly record of life as lived in the first 4 weeks following LSFS from the patient perspective

2. To understand the immediate post-operative recovery experiences of patients and capture the strategies and coping mechanism patients employ during their recovery.

# Materials and methods

## Design

A qualitative study was designed using an Interpretative Phenomenological Analysis (IPA) approach and the study protocol published [19]. IPA informed data collection and data analysis, following a hermeneutic or interpretative pathway informed by Heidegger's perspective of phenomenology [23]; capturing a participant's lived experience during their personal journey through recovery following LSFS. To ensure rigor and comprehensiveness of the findings, the study is reported according to the Standards for Reporting Qualitative Research, a generic checklist for qualitative research [24] and the principles from the Consolidated Criteria for Reporting Qualitative Studies [25]. This study is part of a wider study aiming to understand patients' entire journeys from diagnosis, to surgery and through recovery / rehabilitation.

## Data collection method

Participants completed a diary (S1 Appendix) expressed in their own words capturing a weekly record of their life as lived. The diary questions were developed and piloted, from analysis of the existing literature and through Patient and Public Involvement (detailed later). Minor

changes for clarity of wording were subsequently made. The diary consisted of 12 sections: progress, recovery, motivation, symptoms, medications, healthcare appointment, rehabilitation, positive/negative thoughts, and significant moment during the past week and a comparison to the previous week. The final sections enabled participants to record how they had felt over the previous week or their specific experiences.

## Participants

A purposive sample of 43 adult participants aged ≥16 years undergoing up to 4 level LSFS for back pain and/or leg pain from a degenerative cause (including isthmic spondylolisthesis) at 4 UK spinal surgery centres were recruited to ensure patterns of similarities and differences in their individual journeys were explored. Exclusion criteria included revision surgery or LSFS for traumatic or pathological fracture, infection, malignancy or deformity. The purposive sampling strategy ensured a range of participant characteristics were captured including: indication for surgery, age, gender, ethnicity [17] and factors predictive of outcome such as psychological factors, educational background and number of fused levels [26]. This strategy ensured that the wide clinical heterogeneity of the LSFS population and their outcomes were reflected in the study population.

## Recruitment strategy

Potential participants were identified by a clinical site lead in discussion with the surgeon and their respective team. This process took place in the outpatient clinic at the point where decision for elective surgery was made. Posters in the clinics served as a reminder to the clinical team of the study and encouraged patients to raise questions about the study. The site lead monitored booking lists of patients listed for LSFS and discussed with the booking clerk any potential participants meeting eligibility criteria. At approximately 6 weeks prior to their surgery patients attended the hospital and were consented for LSFS, and a tentative date for surgery was provided. At this appointment, the study was introduced and patients were given the Participant Information Sheet and asked if they would be willing to be contacted further about the study. Patients were afforded the opportunity to ask any questions of the site lead or recruiting research nurse. The site lead/research nurse contacted interested patients by telephone two weeks prior to surgery to discuss any questions they may have regarding the study and to see if they were still interested in participating. Written informed consent to participate in the study was obtained prior to surgery according to patient preference (patients also had the option to consent post-surgery).

## Introduction of patient diary

The diary was introduced and explained by the research nurse/site lead at recruitment to enable participants to familiarise themselves in advance of the surgery with the content and process involved. This also enabled the answering of any questions relating to the diary and the opportunity to discuss the participant's preference for diary format.

## Data collection

To maximise completion and data quality, participants were given the option to complete the diaries in paper format, word document, online via a survey tool or audio recorded. Strategies to enhance diary adherence included a weekly electronic (text or email) or telephone prompt from the research team (University of Birmingham) according to preference, to remind completion and monthly diary collection (post or email). A small financial incentive of £25 was

provided to all participants for the completion of all diary entries (over the entire 12 months of the wider study).

## Ethical approval

Ethical approval was obtained from the South Central—Oxford A Research Ethics Committee (REC ref: 17/SC/0311).

## Data storage and management

All investigators and study site staff adhered to the requirements of the Data Protection Act 2018 with regards to the collection, storage, processing and disclosure of personal information and upheld the Act's core principles. Personal information were collected and stored electronically on a password-protected computer at the University of Birmingham. Data were coded, and depersonalised data with the participant's identifying information were replaced by an unrelated sequence of characters. Secure maintenance of the data ensured that the linking code were kept securely in a separate location using encrypted digital files within password-protected folders and storage media. Only the Chief Investigator and the study Research Fellow had access to the data as necessary for quality control, audit and analysis. After completion of the study, data will be preserved and accessible for 10 years only to the research team.

## Data analysis

All completed diaries were de-identified, transcribed verbatim and coded in accordance with IPA [23]. A framework analysis for individual diaries, multiple diaries for the same participant and then across participants (both deductive and inductive component) was carried out by coding sections of the diaries. Qualitative analysis proceeded in 4 stages.

**Stage 1.**   Researcher FJ transcribed the diary texts verbatim. Blind researchers AR and AS checked the transcribed texts and codes for accuracy. Texts were coded both inductively and deductively capturing the sensitising concepts of positive and negative experiences of participants during their recovery by FJ.

**Stage 2.**   Preliminary themes were presented to researchers AR and AS for discussion. Data were coded in accordance with IPA [23]. Themes and sub-themes were scrutinised until a consensus was reached and researchers were satisfied that the data were representative of the diary texts and displayed in a meaningful way.

**Stages 3 and 4.**   Codes were grouped into themes in an iterative process and tabulated in a summary table (S2 Appendix), illustrated by verbatim extracts. These themes were then checked for consistency and accuracy by researchers AS and AR and discussion with the steering group, including Patient and Public Involvement representatives. Throughout, researchers AR and AS discussed and critiqued *a priori* concepts with FJ and how this may have shaped the themes.

## Transparency and trustworthiness of findings

To ensure trustworthiness and transparency, the data were rigorously checked and re-checked to the level of minor themes. Coding was discussed throughout an iterative process with two experienced researchers (AR and AS) to achieve the double hermeneutic required by IPA [27]. Data were constantly compared and discussed to achieve the ideography required by IPA [27] where individual unique cases are considered first before looking for convergent and divergent details. Furthermore, all investigators discussed their preconceived beliefs openly and

acknowledged their potential impact on the data. Engaging in frequent reflexive discussions with other researchers allowed for greater transparency of the data presented.

### Patient and Public Involvement

Patients who have experienced LSFS were part of the research team, working as co-investigators to ensure the patient perspective was central to this study. Patients contributed to the development of the Participant Information Sheet, consent form and diary format and content. They contributed importantly to the processes of data analysis and interpretation. Researcher NH led on Patient and Public Involvement to ensure that training, support and involvement was always appropriate. The Guidance for Reporting Involvement of Patients and the Public (version 2 short form, GRIPP-2SF) was used to report involvement [28].

## Findings

### Participants

Forty-three participants were recruited pre LSFS. Of these, n = 12 participants withdrew following surgery feeling that they could not now commit to participate; and 3 participants did not proceed to surgery. Twenty-eight participants therefore completed the diaries. The diary completion rate was 89.8%; with n = 19 participants returning 4 completed diaries (1 diary entry per week), n = 7 returning 3 completed diaries, and n = 2 returning 2 completed diaries. Table 1 displays participant demographics, with all information gained from participants. Most participants (n = 18, 64.3%) were aged between 45–64 years, and there was an equal number of male and female participants. Interestingly, numerous participants were unaware

**Table 1. Participant demographics (n = 28).**

|  |  | N (%) |
|---|---|---|
| Age, years | 25–34 | 2 (7.1) |
|  | 35–44 | 3 (10.7) |
|  | 45–54 | 10 (35.7) |
|  | 55–64 | 8 (28.6) |
|  | 65–74 | 4 (14.3) |
|  | ≥75 | 1 (3.6) |
| Gender | Female | 14 (50.0) |
|  | Male | 14 (50.0) |
| Participants perception of reason for surgery | Degenerative | 9 (32.1) |
|  | Spondylolisthesis | 7 (25.0) |
|  | Unknown | 12 (42.9) |
| Participants understanding of number of levels fused | 1 | 14 (50.0) |
|  | 2 | 1 (3.6) |
|  | 3 | 1 (3.6) |
|  | Unknown | 12 (42.9) |
| Work Status | Retired | 4 (14.3) |
|  | Sick leave company | 7 (25.0) |
|  | Sick leave statutory | 5 (17.9) |
|  | Unemployed (≥6 months) | 3 (10.7) |
|  | Other | 6 (21.4) |
|  | Unknown | 3 (10.7) |

of their reason for surgery (n = 12) or detail of the number of levels that they had had fused during the procedure (n = 12).

## Emergence of recovery trajectories

From discussion between researchers during initial coding of the data and identification of preliminary themes, it was clear that participants were following different pathways. The phrase recovery trajectories was consequently used to capture the interplay between the construct of recovery and the participants' experiences of LSFS and their capacity to manage and live with impairments and their subjective experiences. Exploration of the lived experiences from the perspective of those adjusting to and living with the effects of surgery facilitated the identification of the 3 distinct recovery trajectories that emerged from the data: "meaningful recovery", "progressive recovery" and "disruptive recovery". Participants described multiple factors that influenced their construct of 'recovery' and their particular trajectory. This led to a deeper exploration of recovery trajectories evident in participants' accounts and analysis of the inter-related factors that participants perceived as having influenced these trajectories. Important differences in the pace and experiences of recovery were noted by individual participants. These differences (factors influencing the construct of recovery) shaped the recovery trajectories for the individual participant. Fig 1 provides an illustration of the 3 trajectories and the factors influencing the construct of recovery.

## Trajectory of meaningful recovery

A trajectory of meaningful recovery was characterised by n = 10 participants experiencing the effect of their surgery and related impairments associated with their surgery initially as disruptive, in terms of the implication this may have on their social identity and its consequences for

**Meaningful recovery (n=10)**
Managing impairment
Ability to return to ADL
Re-establishing roles and relationships
Reclaiming physical and functional ability
Awareness of progress

**Factors influencing construct**
- Pain
- Physical and functional ability
- Support
- Movement
- Activities of daily living
- Social engagement
- Self-efficacy

**Progressive recovery (n=10)**
Impaired physical and functional ability
Slow return to ADL
Pacing difficulties
Reliance on family support
Reliance on equipment
Slow return to social activities

**Disruptive recovery (n=8)**
Ongoing pain
Ongoing family support
Unable to resume ADL
Unable to engage in physical and functional activities

**Fig 1. Recovery trajectory following lumber spinal fusion surgery.**

their daily lives. The most common description was re-establishing a way of being at home, engaging in meaningful occupation and reconnecting with family members. These factors were important for participants, as was not feeling judged because of their impairments but feeling understood, accepted and taken seriously, with credibility of their symptoms. The immediate disruption was then followed by a process of gradual recovery in which participants endeavoured to engage in physical and functional activities that needed to be achieved in order to return to their functionality and mobility that existed before surgery or before beginning of symptoms. The ability to do this meant actively reducing the disruption of the surgery on their daily activities and social identities that had value to them.

Participants associated meaningful recovery to physical and functional improvement and managing their impairment (physical/functional) with increasing confidence. Participants alluded to their prior level of physical and functional capabilities as a benchmark. Participants within this trajectory would typically break down their diary entries into the effects of surgery (including the experience of pain, limitations in movement), documentation of meaningful improvements (identification of changes), and re-establishing roles.

Participants described their immediate post-discharge experience following LSFS. Their account relates to the difficulties and challenges that their physical and functional impairments had on their daily lives in the first four weeks of their recovery as illustrated by the following diary extracts.

> *"Fears of reoccurrence of pre-surgery symptoms. Am I going to be able to do what I did before?"* [P5]

> *"Don't feel that much progress has been made—pain is still a major factor in being able to do much"* [P9]

Participants illustrated a range of themes that characterised this trajectory.

**Managing impairment.** Participants described their physical and functional impairments immediately following discharge and transitioning home. In the first few weeks following discharge, participants noted small but meaningful improvements largely due to a desire to get better and resume their normal activities. Some participants reported recovery as slower than anticipated.

> *"I am definitely improving with regard to mobilising and other daily activities. Nerves are slow to heal, but they can heal"* [P5]

> *"Motivation is still high. Pushing myself past what I was doing last week but obviously still taking it slowly. But progress is going in the right direction"* [P4]

**Ability to return to ADL.** Participants appeared to base their recovery on their ability to carry out routine functional activities. Participants referred to activities of daily living as routine aspects of self-care such as dressing, bathing, cooking, housework and shopping. It was when attempting to complete these tasks that participants noticed experienced difficulties following surgery.

> *"Physically trying to extend the walks, able to do more household tasks and feeling I am contributing to the household"* [P5]

> *"More achievement. Can put on and take off compression stocking myself so can have a shower whenever I want. Can load/unload hang out the washing without help. Wanted to be*

*driver to town (1 mile) to do some shopping. Went to a farm shop for tea and had a nice drive round country side" [P11]*

**Re-establishing roles and relationships.**   Pain and immobility affected participants' abilities to engage in activities of daily living, but they described subtle and meaningful improvements in their mobility and pain with the assistance of family and friends that contributed to them re-establishing roles and relationships with their family, friends and wider community. This was the first stage of re-establishing role and relationships.

*"I need to get better to show my friends and family that their support has assisted me with the process" [P5]*

*"After speaking to family and friends I have come to see that my progress so far is positive" [P1]*

**Reclaiming physical and functional ability.**   Participants described their progressive engagement in a range of physical and functional activities to ensure a speedy recovery. An important focus for participants was recovering from surgery and getting back to the level of functionality and mobility they were at prior to the onset of low back pain–their benchmark.

*"This week has been extremely good, walking further and climbing stairs as normal. Been shopping and out for meals so onwards and upwards" [P4]*

*"Walking without pain, being able to stand up straight. Getting back to normal" [P24]*

**Awareness of progress.**   Participants' desires to gradually regain a level of independence, namely the physical and functional control which they enjoyed prior to the onset of low back pain were captured. A key factor that contributed to participants regaining normality and independence was their awareness of the consistent and progressive improvement in their physical and functional abilities and pain management.

*"This week has resulted in me becoming much more independent with regard to the daily activities of living, with some obvious restrictions with regard to food preparation. I am assisting with daily household duties as far as my surgery currently allows, which in turn helps to foster feelings of positivity"[P5]*

*"Realistic pain levels due to personal decision to reduce medication" [P3]*

## Trajectory of progressive recovery

A trajectory of progressive recovery was characterised by n = 10 participants experiencing an initial period of disruption/adjustment followed by a focus on small but meaningful improvements in their physical and functional abilities. Participants described having to focus on their changed/limited physical and functional ability post-surgery, which became starkly apparent following discharge from hospital and recuperating at home. Over time, participants gradually noticed small but meaningful improvement as their physical ability improved and they began to feel more confident engaging in routine daily activities including working towards further improvement in their mobility. Participants illustrated a range of themes that characterised this trajectory.

**Impaired physical and functional ability.**   For some participants the challenges and fluctuations they faced with their physical and functional ability had serious bearing for their recovery. They initially held the assumption that their level of physical and functional abilities

would be promptly restored, subsequently realising that recovery may be a lengthy process and would not meet the expectations initially set.

*"My level of expectation to walk I would be at able to walk was 60%/70% recovered. I am approximately half even less at 25%" [P12]*

*"I appreciate it is going to be a long rehabilitation process and I am slowly accepting that fact. Recovery is slow and steady and I maybe hadn't quite prepared myself for that" [P52]*

**Slow return to activities of daily living (ADL).** Participants described needing a great deal of assistance with ADL during the first week following surgery. Participants referred to ADL as basic self-tasks such as washing, dressing, housework and shopping. It was when attempting to undertake these activities that participants noticed the struggle they were now experiencing since their surgery.

*"Mixture of emotions. Ranging from euphoria to anxiety. Lots of feelings about impact and outcomes of surgery on activities of daily living and work commitments" [P5]*

*"Definitely frustrated. I know this is going to take a long time to recover from, but I'm an active guy and not being able to do the most basic things like do the washing up, or shower without excruciating pain is infuriating" [P26]*

**Pacing difficulties.** Most participants recognised the need to adopt alternative strategies to undertake routine functional activities, notably in the early days following surgery. Participants reported using bath boards for showering and taking frequent rests after pacing activities. These strategies were still reported at 4 weeks following surgery, as was the requirement to pace and take time to carry out activities including household chores and gardening.

*"Recovering is making slow progress though I am able to do more things and exercise" [P19]*

*"Continue and extend walking do more chores around house e.g. laundry" [P11]*

**Reliance on family support.** Participants recognised needing substantial support from family/friends immediately following discharge and throughout the first 4 weeks of their recovery. Receiving support was closely related to a participant's ability to manage their post-operative pain, ADL and movement.

*"Mild degree of frustration with regard to not feeling able to undertake household chores. I feel grateful and blessed that I have a fantastic family and friends who are on hand to assist me as necessary" [P5]*

*"I would like to be able to do some of the simplest things for myself instead of relying on my wife to do as she suffers from Ankylosing Spondylitis" [P14]*

**Reliance on equipment.** Participants described the use and their reliance on walking aids and toilet equipment as vital to maintain and boost confidence, offer reassurance, prevent falls and offer support during the most vulnerable period of their recovery.

*"Still maintaining structure to my day & ensured I have got up & dressed. Bath board arrived from equipment services which made quite a difference—such a mood boost to be able to get showered properly" [P22]*

*"Think I could do with a few aids at home to help e.g. the toilet is too low and painful to sit down on. Also once I've got sat in bed I haven't anything to pull myself to standing position which is also painful and stressful" [P8]*

**Slow return to social activities.** Participants described their ability to socialise and leave their home as an important milestone. However, this was fraught with challenges, as some participants described the need to devise strategies to overcome their anxiety and environmental barriers such as physically managing to leave the house.

*"Travelling in a car and waiting for the ceremony to start in a public place. Slight feeling of vulnerability but glad to be outside and back in the real world" [P5]*

*"Can journey to Bromsgrove to visit friends- went well more comfortable than expected" [P19]*

## Trajectory of disruptive recovery

A trajectory of disruptive recovery was characterised by n = 8 participants experiencing sustained disruption and a continued sense of struggle towards any meaningful recovery. Participants related the same meaning and consequences as the two previous trajectories, but the difference was that they perceived no sense of working towards any meaningful recovery. On the contrary, their struggle and suffering continued and their engagement with any meaningful recovery plan was limited. Participants illustrated a range of themes that characterised this trajectory.

**Ongoing pain.** Participants reported varying levels of continued pain, and for a significant number of participants pain continued to improve following discharged. Those who struggled with pain immediately after discharge continued to experience pain and for some their pain experience had deteriorated.

*"I have had chills and sweats, severe pain and tiredness at times through the week" [P1]*

*"The most notable and hideously painful full back spasm episode. This type of spasm has the ability to completely immobilise me and is quite frightening to be honest" [P5]*

Constipation was a common problem for many participants, frequently causing considerable discomfort and pain.

*"Sharp pain in my left side and buttock, bladder weakness and constipation [P8]"*

*"Nothing really negative, you can't get over having no leg pain. Constipation was the worst thing [P24]"*

Some participants were worried about wound infection and its impact on their recovery.

*"Pain in my back. Got an infection in my wound and went to the walk-in centre, they cleaned it and gave antibiotics. Left hospital numb and tingly right top leg, it's still there [P3]"*

*"Very bad, lots of worry about the level of infection and how it could affect me [P21]"*

**Ongoing family support.** Participants described seeking support from family and friends with their recovery. Support was characterised as getting direct help, for example with household chores, from family and friends. The need for support was closely linked with

participants' requiring assistance with ADL and struggling to develop the necessary motivation to participate in rehabilitation and willingness to make lifestyle changes.

> *"I am assisting with daily household duties as far as my surgery currently allows, which in turn helps to foster feelings of positivity. Mild degree of frustration with regard to not feeling able to undertake household chores. I feel grateful and blessed that I have a fantastic family and friends who are on hand to assist me as necessary"* [P5]

> *"When I woke up on Saturday I just wanted to scream and let all the frustration out. I think I was trying to tidy up my bedroom, but in the end I needed to call a friend and she stopped me from spiralling down into tears"* [P26]

**Unable to resume ADL.**    Participants appeared to base their recovery on their ability to perform basic ADL following discharge from hospital and recovering at home. Participants often described the most important ADLs as being able to shower independently, to get dressed and requiring minimal assistance going to the toilet. Participants described being unable to perform basic self-care tasks and resume normal activities.

> *"Painful at night, finding it hard to get to sleep gone midnight. Hard in shower need help"* [P2]

**Unable to engage in physical and functional activities.**    Participants reported the detrimental effect the surgery had on their physical and functional capabilities. Participants reflected on their inability to fully engage in physical and functional activities and the extent to which these were affected, and expressed their apprehension about returning to normal physical and functional activities following surgery.

> *"I am doing very little exercises due to back pains"* [P1]

> *"As in the past the lack of a specific person to contact means that with a physio/exercise problem I must wait 2 weeks for a resolution and as such this relates to 2 weeks lost recovery and an increase in personal frustration"* [P23]

### Factors influencing the construct of recovery

The frequencies of occurrence of factors influencing the construct of recovery to shape the recovery trajectories for the individual participant are illustrated in Fig 2.

**Pain.**    The different trajectories of recovery and diversity of experience identified were influenced by a number of factors. Pain was a significant factor influencing the recovery pathway for many participants and was related to outcome, for example, ADL and physical and functional abilities. Participants related pain to anxiety in general.

> *"Pain is limiting my ability to move forward as much as I like"* [P9]

> *"The realisation that a different type of pain / discomfort would need to be dealt with post operatively in order to continue making progress. Balancing pain medication relief is significant as it impacts on everything"* [P5]

> *"The pain reduced me to tears, but this could have been coupled with anxiety. Luckily my husband was around- not that he could help in any way- it had to pass itself"* [P5]

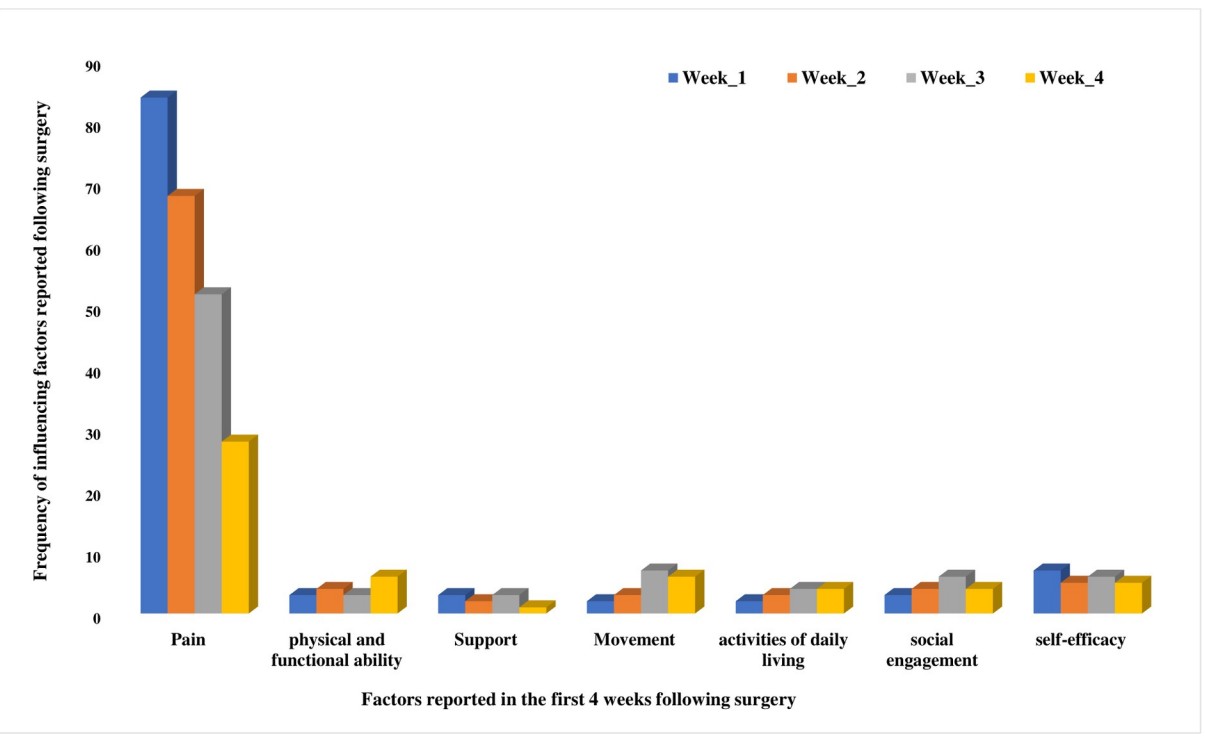

**Fig 2. Factors influencing construct reported in the first 4 weeks following Lumber Spinal Fusion Surgery (LSFS).**

**Physical and functional ability.**   Participants shared valuable insight into how their expectations for their recovery period and the role of physical and functional ability had influenced their ability to return to their pre-surgery activities.

*"Physically trying to extend the walks, able to do more household tasks and feeling I am contributing to the household"* [P5]

*"So can have a shower whenever I want. Can load/unload hang out the washing without help. Wanted to be driver to town (1 mile) to do some shopping. Went to a farm shop for tea and had a nice drive round country side"* [P11]

**Support.**   Most participants mentioned seeking help from family and friends at some point during their recovery, and requiring assistance highlighted their limitations compared to before the surgery. Participants alluded to some concerns regarding the stress and physical demands they were inadvertently putting on those supporting them.

*"I am very blessed and grateful. And I need to get better to show my friends and family that their support has assisted me with the process"* [P5]

*"It's still to be able to help the wife and take some pressure off her with doing household things"* [P5]

**Movement.**   Participants felt inhibited and considered their surgery as an obstacle towards the lack of function and walking ability that they experienced immediately after surgery.

*"Stairs and walk around house very slowly, not a lot at once"* [p2]

*"I managed to use a walking frame to walk a few yards on 11th Oct. and walked down the corridor and a few steps using crutches on 12th. On 13th & 14th I was at home were although I felt I had left hospital a bit early I did manage to get around the house and use the stairs using my crutches"* [p1]

**ADL.** Quality of life was strongly correlated with activities of daily living. Quality of life was often poorer among those who could not perform basic activities of daily living. Immediately after surgery the level of support needed sharply increased for many more daily tasks.

*"There have been varying times when I have woken/got out of bed, but have ensured that I freshened up & got dressed rather than sit in pyjamas all day. (Unable to shower as still waiting for bath board to be delivered by equipment services"* [P22]

*"Able to carry out exercises and activities easier tidied out hypes and did a little cooking. Not yet back to hobbies"* [P6]

**Social engagement.** Social participation was viewed as a lifelong engagement process and the ability to engage and maintain activities that bring quality to participant's life post-surgery in view of resuming a normal life after surgery. Participants related social engagement to feelings of depression.

*"The feeling of being a failure and a burden, as still not driving. Suffer from depression and have been in a black hole. It doesn't help as my work friend is away on holiday"* [P25]

*"I've been fit enough to host friends and play with my niece and nephews. I'm tired"* [P26]

**Self-efficacy.** Belief in self-efficacy and towards their recovery influenced a participant's ability to take control of their recovery, and to adapt and manage their recovery journey. Alternatively, low self-efficacy appeared to interfere with recovery.

*"Motivated to walk about because I know it helps recovery"* [P3]

*"Sometimes think I'm not going to make a full recovery and I'm still going to have a back problem"* [P1]

## Discussion

This is the first study to explore patients' lived experiences in the immediate period following LSFS through a weekly record in real time. The IPA approach enabled new insights by understanding how patients make sense of their immediate recovery period. The findings uncovered aspects of the recovery journey that have not been previously identified in other studies, in particular the 3 trajectories.

LSFS is widely used for people experiencing severe, life-limiting, chronic low back pain that does not respond to conservative treatments [5], and it is often therefore the last resort. It was interesting that n = 3 participants recruited pre-surgery (consented for surgery) did not go on to have the surgery, illustrating the ongoing consideration of indications for surgery within spinal centres.

In depth examination of the patient diaries identified 3 distinct recovery trajectories within participants' diverse and vivid descriptions of their recovery experiences: meaningful recovery (engagement in physical and functional activities to return to functionality/mobility); progressive recovery (small but meaningful improvement in physical ability with increasing

confidence); and disruptive recovery (limited purpose for meaningful recovery). Recovery from LSFS has not been described in the literature as a linear process, often referring to physical and functional rehabilitation and adjustment in the first few months following surgery [29, 30]. Our findings support this and show that recovery is a complex and dynamic process often interrelated between physical, functional, social and emotional domains. In the meaningful recovery trajectory participants described a process of gradual engagement to regain their level of mobility and functionality experienced prior to their low back problem; contrasting to the other extreme, where participants in the disruptive trajectory experienced sustained disruption and a continued sense of struggle towards any meaningful recovery. Between these extremes, participants following a progressive recovery, experienced small but meaningful improvements in their physical and functional abilities to enable them to progressively engage more actively in routine ADL.

The variation in participants' experiences can be traced to the recovery trajectory participants adopted within the first 1–2 weeks following surgery. Our data suggest that immediate recovery was generally a difficult and painful period. Participants experienced debilitating symptoms, including severe pain, constipation, fatigue and anxiety. The meaning of their surgery and subsequent recovery was shaped by their ability to adapt and manage the speed of their recovery, by acknowledging their limitations but also by taking advantage of the help and support available to them. For example, presence of pre-operative symptoms was a strong stimulus through the factor of pain influencing the construct of recovery. Pain impacted on the speed of recovery and for some participants their sustained pre-operative pain caused a lengthy and slow recovery. This is consistent with previous data from the Swedish National Spine Register that highlighted 25% of patients reporting no change or worsened pain following surgery [14].

Participants who recorded a negative 'most memorable moment' (e.g. scared to return home) during the initial week post-surgery reported this consistently in subsequent weeks and were characterised by the disruptive recovery trajectory and low self-efficacy. An example of a negative early most memorable moment is:

> *"It hurts and I feel unsupported by the hospital now I'm home (I do have the physio team's helpline to call if needed). Think I could do with a few aids at home to help e.g. the toilet is too low and painful to sit down on. Also once I've got sat in bed I haven't anything to pull myself to a standing position which is also painful and stressful" [P8]*

This is consistent with previous studies that have shown that patients with a positive outlook on their recovery appear to be more resilient and have better physical and functional outcomes [31]. It is also consistent with the placebo literature [32], and a narrative synthesis informing development of a framework that identified the importance of meaningful activities, experiences and interactions to establishment of hope following a stroke [33].

Our findings support those from Abbott et al [20] identifying that the issues important to patients relate to the body functions and structures, activities and participation, and environmental components of the ICF; in particular the factors influencing the construct of recovery. For example, findings highlight the importance of relationships, support and social interaction as important factors in shaping participants' trajectories. The factors identified are consistent with those influencing hope in spinal cord injury, for example social isolation and a lack of social support can influence patient outcomes [34]. In line with Abbott et al [20], these findings support that these qualitative data must inform the evaluation of patient outcomes following LSFS to determine outcomes of importance to patients.

### Further research

It will be interesting to see if trajectory is influenced by the pre-operative or operative experience and we will be able to evaluate this through analysis of in-depth semi-structured interviews in the first 2 weeks following surgery as part of the wider study [19]. It will also be valuable to see if patients remain on the same trajectory throughout their longer term recovery, and whether the factors influencing the construct of recovery change over the longer term; particularly as evidence of long-term outcome is limited [13], and at 12 months following surgery 40% of patients report dissatisfaction with outcome [14]. The wider study will enable us to look at 12 months of diary data and interviews at 12 months [19]. The findings are also important to influence patients' expectations of LSFS and these findings can inform what patients are told prior to surgery regarding what they should expect from surgery.

### Strengths and limitations of the study

Patient diaries were an effective method for collecting a rich density of data, providing a record of events throughout a participant's recovery using a narrative account of their day-to-day activities in their own environment. Diary adherence was very good and it was therefore an acceptable method to participants. This study has a key limitation as although participants were purposively recruited to reflect a range of characteristics, the participants were entirely identified as 'White British', limiting transferability of findings. Interestingly, n = 12 participants withdrew following surgery feeling that they could not participate in completing the diaries having experienced the surgery. Unfortunately, no further data regarding these patients were available.

## Conclusion

This is the first account of immediate recovery trajectories from patients' perspectives. The findings show that surgery impacts all aspects of a person's life, and that recovery is not just rehabilitation of physical and functional abilities. Recovery is a complex and dynamic process comprising of the interrelationship between physical, functional, emotional and social factors [35]; described through 3 distinct trajectories of 'meaningful recovery', 'progressive recovery' and 'disruptive recovery'. Recognition of a patient's trajectory through their examination of the patient including relevant patient reported outcome measures, may inform patient-centred recovery, pre-operative preparation (information provided including what patients should expect e.g. levels of pain, timing to return to function etc), follow-up and rehabilitation to improve patient outcomes.

## Supporting information

**S1 Appendix. Sample diary.**
(PDF)

**S2 Appendix. Description of themes.**
(PDF)

## Acknowledgments

Eleanor Keeling and Sarah Rich from the Royal Orthopaedic Hospital.

Jayne Edwards, Claire Wright and Lisa Burgess Collins from the Robert Jones and Agnes Hunt Orthopaedic Hospital NHS Foundation Trust, UK.

Tim Noblet, Michael Dunn and Abby Newdick from St George's University Hospitals NHS Foundation Trust, UK.

Julie Sterling and Marie Morley, patient involvement.

## Author Contributions

**Conceptualization:** Alison Rushton, J. Bart Staal, Martin L. Verra, Andrew Emms, Michael Reddington, Ashley Cole, Paul C. Willems, Lorin Benneker, Nicola R. Heneghan, Andrew Soundy.

**Data curation:** Alison Rushton, Feroz Jadhakhan, Annabel Masson, Victoria Athey.

**Formal analysis:** Alison Rushton, Feroz Jadhakhan, Annabel Masson, Victoria Athey, Andrew Soundy.

**Funding acquisition:** Alison Rushton, J. Bart Staal, Martin L. Verra, Andrew Emms, Ashley Cole, Paul C. Willems, Lorin Benneker, Nicola R. Heneghan, Andrew Soundy.

**Investigation:** Alison Rushton, Annabel Masson, Andrew Emms, Michael Reddington, Ashley Cole, Andrew Soundy.

**Methodology:** Alison Rushton, J. Bart Staal, Martin L. Verra, Michael Reddington, Paul C. Willems, Lorin Benneker, Nicola R. Heneghan, Andrew Soundy.

**Project administration:** Alison Rushton, Annabel Masson, Nicola R. Heneghan.

**Resources:** Alison Rushton.

**Supervision:** Alison Rushton, Nicola R. Heneghan, Andrew Soundy.

**Validation:** Alison Rushton.

**Visualization:** Alison Rushton, J. Bart Staal, Martin L. Verra, Andrew Emms, Michael Reddington, Nicola R. Heneghan, Andrew Soundy.

**Writing – original draft:** Alison Rushton, Feroz Jadhakhan, Andrew Soundy.

**Writing – review & editing:** Alison Rushton, Feroz Jadhakhan, Annabel Masson, Victoria Athey, J. Bart Staal, Martin L. Verra, Andrew Emms, Michael Reddington, Ashley Cole, Paul C. Willems, Lorin Benneker, Nicola R. Heneghan, Andrew Soundy.

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
