## [Decision Letter · Decision Letter 0]

4 Sep 2020

PONE-D-20-20781

Patient journey following lumbar spinal fusion surgery (FuJourn):  a multicentre exploration of the immediate post-operative period using qualitative patient diaries

PLOS ONE

Dear Dr. Rushton,

Thank you for submitting your manuscript to PLOS ONE. After careful consideration, we feel that it has merit but does not fully meet PLOS ONE’s publication criteria as it currently stands. Therefore, we invite you to submit a revised version of the manuscript that addresses the points raised during the review process.

We look forward to receiving your revised manuscript.

Kind regards,

Panagiotis Kerezoudis, M.D., M.S.

Academic Editor

PLOS ONE

Journal Requirements:

Reviewers' comments:

Reviewer's Responses to Questions

**Comments to the Author**

1. Is the manuscript technically sound, and do the data support the conclusions?

Reviewer #1: Yes

Reviewer #2: Partly

Reviewer #3: Yes

2. Has the statistical analysis been performed appropriately and rigorously? 

Reviewer #1: N/A

Reviewer #2: N/A

Reviewer #3: No

3. Have the authors made all data underlying the findings in their manuscript fully available?

Reviewer #1: Yes

Reviewer #2: Yes

Reviewer #3: Yes

4. Is the manuscript presented in an intelligible fashion and written in standard English?

Reviewer #1: Yes

Reviewer #2: Yes

Reviewer #3: Yes

5. Review Comments to the Author

Reviewer #1: The authors present a unique study of 28 patients who contributed weekly diaries in the first four weeks following short construct lumbar fusion. They identified three distinct trajectories of "meaningful recovery, progressive recovery, and disruptive recovery." This is an interesting study that may help with patient education of what to expect in the immediate postop period. Of course, no conclusions can be made about the efficacy of lumbar spine fusion surgery from this study as full recovery takes at least 3-6 months and only at that point can we start to determine if surgery was a success.

1) Were there any differences in the patient population, pre-op opioid use, indications for surgery, or technical aspects of the operation that could account for differences in short term recovery trajectory?

2) An interesting addition to the study or future study, as the authors have already indicated, would include a final diary entry at long term follow up to reflect on the entire process and see how the long term perception of the postoperative course compares to the short term.

3) It would also be interesting to look at traditional PRO measures such as ODI and see how that correlates with the "construct of recovery".

Reviewer #2: This article attempts to track patient physical, psychological, and social factors and correlate them with patient recovery. The 3 types of recovery meaningful, progressive, and disruptive with typical features are correlated with patient’s diaries of post-operative experiences. This allows for grouping of patient’s under each type.

The study has multiple opportunities to add semi-objective/objective data. By this, I mean mobility monitoring, pain scale/FACES, and functionality scale used to objectively assess patient recovery. This will add more credibility to the paper. The IPA approach is good for diary analysis/grouping however is not sufficient alone.

Further, the overall study lack external validity which is discussed within the paper. The entire population consists of white UK citizens.

This paper is unsalvageable. I recommend the addition of the above comments and more diverse sampling.

Reviewer #3: This study is very well done however I would like to see one change, I would like a chart representing the parameters you outlined which were in their journals (pain, recovery, family support etc) showing how many patients filled each category in order to summarize it all. By doing this it will be clearer to the reader how many patients of the ones with good outcomes fulfilled each category which can lead to further studies.

---

## [Author Response · Author response to Decision Letter 0]

7 Oct 2020

Document attached with detailed response to reviewers' feedback.

---

## [Decision Letter · Decision Letter 1]

23 Oct 2020

Patient journey following lumbar spinal fusion surgery (FuJourn):  a multicentre exploration of the immediate post-operative period using qualitative patient diaries

PONE-D-20-20781R1

Dear Dr. Rushton,

We’re pleased to inform you that your manuscript has been judged scientifically suitable for publication and will be formally accepted for publication once it meets all outstanding technical requirements.

Kind regards,

Panagiotis Kerezoudis, M.D., M.S.

Academic Editor

PLOS ONE

Additional Editor Comments (optional):

The authors have addressed the reviewers' comments and the manuscript is now ready for publication.

Reviewers' comments:

Reviewer's Responses to Questions

**Comments to the Author**

1. If the authors have adequately addressed your comments raised in a previous round of review and you feel that this manuscript is now acceptable for publication, you may indicate that here to bypass the “Comments to the Author” section, enter your conflict of interest statement in the “Confidential to Editor” section, and submit your "Accept" recommendation.

Reviewer #1: All comments have been addressed

2. Is the manuscript technically sound, and do the data support the conclusions?

Reviewer #1: Yes

3. Has the statistical analysis been performed appropriately and rigorously? 

Reviewer #1: Yes

4. Have the authors made all data underlying the findings in their manuscript fully available?

Reviewer #1: Yes

5. Is the manuscript presented in an intelligible fashion and written in standard English?

Reviewer #1: Yes

6. Review Comments to the Author

Reviewer #1: No new comments. It appears that they addressed my questions with the current manuscript or with future studies.

7. PLOS authors have the option to publish the peer review history of their article (what does this mean?). If published, this will include your full peer review and any attached files.

Reviewer #1: No

---

## [Editor Report · Acceptance letter]

18 Nov 2020

PONE-D-20-20781R1 

Patient journey following lumbar spinal fusion surgery (FuJourn):  a multicentre exploration of the immediate post-operative period using qualitative patient diaries 

Dear Dr. Rushton:

I'm pleased to inform you that your manuscript has been deemed suitable for publication in PLOS ONE. Congratulations! Your manuscript is now with our production department. 

Kind regards, 

on behalf of

Dr. Panagiotis Kerezoudis 

Academic Editor

PLOS ONE